# Zinc Oxide Nanoparticles Enhance Grain Yield and Nutritional Quality in Rice via Improved Photosynthesis and Zinc Bioavailability

**DOI:** 10.3390/foods14173018

**Published:** 2025-08-28

**Authors:** Jie Chen, Muyan Zhang, Jingtong Sun, Xinyue Liu, Xijun Yuan, Rui Wang, Haipeng Zhang, Yanju Yang

**Affiliations:** Jiangsu Key Laboratory of Crop Cultivation and Physiology/Co-Innovation Center for Modern Production Technology of Grain Crops, Research Institute of Rice Industrial Engineering Technology, Yangzhou University, Yangzhou 225009, China; mz120231333@stu.yzu.edu.cn (J.C.); 231701333@stu.yzu.edu.cn (M.Z.); 243702115@stu.yzu.edu.cn (J.S.); 231702111@stu.yzu.edu.cn (X.L.); mz120231351@stu.yzu.edu.cn (X.Y.); chinaniubi0@gmail.com (R.W.); hpzhang@yzu.edu.cn (H.Z.)

**Keywords:** zinc oxide nanoparticles, rice yield, grain quality, zinc enrichment, foliar fertilization, photosynthetic efficiency

## Abstract

Improving rice yield, eating quality, and zinc (Zn) nutrition is crucial to meet the growing demand for high-quality and nutritious food, while zinc itself plays a pivotal role in rice growth and quality formation. In this two-year field study, we investigated the effects of zinc oxide nanoparticles (ZnO NPs) one-time foliar application at 5, 10, and 20 mg L^−1^ during the gestation stage on grain yield, rice quality, and zinc biofortification. Although some year-to-year variations and year × treatment interaction in the magnitude of these effects were observed, the positive responses to ZnO NPs were consistent across the two years. Results showed that ZnO NPs application increased grain yield by 1.40–4.62%, mainly supported by enhanced net photosynthetic rate and SPAD values after heading. Meanwhile, ZnO NPs significantly enhanced taste value (1.61–7.22%) and breakdown value (5.36–15.63%), while reducing chalkiness rate (5.73–18.62%), chalkiness degree (11.57–27.18%), amylose content (3.72–6.76%), and setback value (8.98–24.53%). Additionally, ZnO NPs markedly increased the zinc content (23.73–85.10% in brown rice; 29.07–103.07% in polished rice) and reduced the phytic acid to zinc molar ratio by 18.46–48.39%, improving zinc enrichment and bioavailability. These findings suggest that ZnO NPs foliar application is effective to simultaneously enhance grain productivity, rice quality, and zinc density.

## 1. Introduction

Rice (*Oryza sativa* L.) is a staple food for over 70% of the global population, playing a central role in ensuring food security and human nutrition, particularly in Asia, Africa, and parts of Latin America [1]. Traditionally, rice breeding and cultivation have focused on maximizing grain yield. However, with the continuous improvement in living standards and consumer health awareness, there is increasing demand for rice with high eating quality and enhanced nutritional value [2,3]. Among various quality and nutritional parameters, zinc (Zn) content in rice grains has gained significant attention due to its crucial role in human health [4].

Zinc is an essential trace element involved in numerous physiological functions, including immune regulation, antioxidant defense, cellular division, and enzyme activation [5]. Zinc deficiency can lead to growth retardation, impaired immune function, skin disorders, and increased risk of chronic diseases [6]. It is estimated that approximately 17% of the global population is at risk of zinc deficiency, particularly in developing countries where diets are dominated by cereal staples with low micronutrient content [7,8]. Since the human body cannot synthesize zinc, it must be obtained through diet, making dietary biofortification of staple crops such as rice a critical strategy to combat global zinc deficiency [9]. Among the approaches for zinc biofortification, fertilizer-based agronomic strategies are recognized as the most direct, economical, and rapidly effective method [10]. Compared with soil application, foliar spraying of zinc fertilizer has been shown to be more efficient in increasing grain zinc concentration in rice. This is because foliar-applied zinc is directly absorbed and transported to developing grains, avoiding zinc fixation and loss in the soil matrix [11,12,13]. Common zinc fertilizers include inorganic salts (e.g., zinc sulfate and zinc chloride), organic zinc complexes, and chelated forms. However, these conventional fertilizers often suffer from limitations such as poor foliar adhesion, low absorption efficiency, potential phytotoxicity, and instability under high-temperature and humid field conditions [14,15,16,17,18].

In recent years, the emergence of nanotechnology has introduced zinc oxide nanoparticles (ZnO NPs) as a promising alternative to conventional zinc fertilizers. ZnO NPs exhibit unique physicochemical properties, including high surface area, excellent stability, improved plant uptake, and low toxicity [19,20]. In particular, compared with conventional zinc fertilizers, nano-zinc oxide can raise zinc-use efficiency from 20–30% to 80–95% [21]. At reduced application rates, it exhibits low phytotoxicity and high zinc bioavailability, and its integration with emerging smart-agriculture technologies such as unmanned aerial vehicles mitigates the risk of zinc toxicity during field application [22]. With future improvements in standardization of safety protocols, cost reduction, and simplification of synthesis, nano-zinc oxide will find even wider application in agriculture [23]. Studies have shown that ZnO NPs can enhance seed germination, chlorophyll synthesis, biomass accumulation, and antioxidant enzyme activity in crops [24,25,26]. Azam et al. found that the soil application and foliar spray of zinc oxide nanoparticles could promote maize growth, increase photosynthetic pigments, and enhance antioxidant activity [27]. Research by Kalal and Jajoo indicated that ZnO NPs could be applied as a promising seed priming agent to improve germination as well as photosynthetic performance of wheat seeds [28]. In rice, Shadma and Singh [29] reported that ZnO NPs improved early growth and yield of rice when applied at the seedling stage. Chutipaijit et al. [30] also demonstrated enhanced growth tolerance and photosynthetic performance in rice following ZnO NPs treatment. Despite these promising findings, most existing research has focused on the effects of ZnO NPs on rice growth and stress resistance, with limited attention given to their impact on rice grain quality and zinc nutritional characteristics. Moreover, the specific influence of ZnO NPs foliar application during critical developmental stages on grain yield, physicochemical properties, and zinc bioavailability remains insufficiently studied. The grain-filling stage after heading is the crucial stage for the formation of yield and quality [31], and foliar spray of zinc fertilizers is commonly chosen around the heading stage, while few studies have delved deeper into the effects on rice grain filling and further on final yield and quality. Therefore, a two-year field experiment was conducted to evaluate the effects of foliar application of ZnO NPs at the gestation stage on grain yield, rice quality, and zinc enrichment in irrigated paddy *japonica* rice. Furthermore, we observed the dynamic changes in photosynthetic rate and SPAD values of leaf photosynthesis after heading to reveal the physiological basis for grain filling and yield and quality formation. This research provides new insights into the application potential of ZnO NPs in the simultaneous improvement of yield, grain quality, and nutritional value in rice biofortification strategies by assessing the yield components, grain physicochemical parameters, zinc bioavailability.

## 2. Methods and Materials

### 2.1. Experimental Site

Field trials were conducted consecutively in 2020 and 2021 on the research farm of Yangzhou University, Jiangsu, China (32°23′24″ N, 119°25′12″ E). The station lies in a rice-wheat rotation belt within the mid-lower Yangtze Basin and experiences a warm, humid subtropical climate (Figure 1). Topsoil (0–20 cm) was a sandy loam with pH 6.51 and contained 24.4 g kg^−1^ organic matter, 1.3 g kg^−1^ total nitrogen (N), 104.2 mg kg^−1^ available N, 35.4 mg kg^−1^ available phosphorus (P), and 72.5 mg kg^−1^ available potassium (K). The fields used for the two-year trial were two plots with similar soil fertility located at the same trial site.

### 2.2. Agronomic Management Systems

Nanjing 46, a regional japonica cultivar (155–160 d growth cycle), was used. Seeds were sown in plastic trays, kept in darkness for 48 h, then transferred to a moist nursery. 25-day-old seedlings were manually transplanted at 1.33 × 10^6^ hills hm^−2^ on 15 June 2020 and 16 June 2021, harvest dates were 30 October 2020 and 31 October 2021, respectively.

A total of 12 plots (10 m × 6 m = 60 m^2^ each) were arranged in a randomized complete block design, with three replicates. Each plot is planted with 1600 holes, with 5 seedlings planted in each hole, for a total of 8000 basic seedlings, ensuring that the number of seedlings in each plot is consistent when transplanted in the field. Using deionized water as a control, three concentrations (5, 10, and 20 mg L^−1^) of zinc oxide nanoparticles (ZnO NPs) were directly applied to rice leaves once without the addition of surfactants at the gestation stage (as the rice spike begins to emerge from the leaf sheath) under calm, clear weather. These treatments were designated T1, T2, and T3. Each treatment was repeated three times, as three plots in the field. ZnO NPs (≥99.99% purity; ≈ 50 nm; specific surface area ≥ 4 m^2^ g^−1^; white powder) were obtained from Shanghai Chaowei Nanotechnology Co., Ltd. Required masses (5, 10, or 20 mg) were dispersed in 1 L ultrapure water via 30 min ultrasonication and foliar sprayed to rice under calm, clear conditions.

Urea (46% N) was split-applied at 270 kg hm^−2^: 30% at transplanting, 30% seven days later (tillering), and 40% at panicle initiation. Calcium superphosphate (135 kg P hm^−2^) and potassium chloride (270 kg K hm^−2^) were incorporated basally. Irrigation, pest, disease, and weed management followed standard regional recommendations.

### 2.3. Sampling and Data Collection

#### 2.3.1. Yield and Yield Components

At maturity, plants within a 1 m^2^ area per plot were hand-cut, threshed, and sun-dried to 14% moisture for yield determination. From each plot, 20 representative hills were selected to record panicle number; a subsample of 12 hills was used to quantify spikelets per panicle, filled-grain rate, and 1000-grain weight.

#### 2.3.2. Net Photosynthetic Rate and SPAD Value

Flag-leaf net photosynthetic rate (Pn) at heading and grain-filling stage was recorded with a Li-6400 portable system (LI-COR, Lincoln, NE, USA). Conditions were set to 1000 μmol m^−2^ s^−1^ light intensity, 400 μmol mol^−1^ CO_2_, 30–35 °C, and 500 μmol s^−1^ flow. The central portion of flag leaves was clamped for 2 min between 09:00 and 12:00 under full sun. Three plants per replicate were assessed.

Flag-leaf SPAD readings were obtained with a SPAD-502 m (Konica Minolta Sensing, Inc., Osaka, Japan). Three positions, as lower quarter, midpoint, and upper quarter of the flag leaves, were averaged per leaf to obtain the representative SPAD value for each treatment.

#### 2.3.3. Determination of Zinc Content and Zinc Distribution in Rice

Dry samples (0.5 g) were transferred to PTFE vessels and digested in 5 mL of ultrapure HNO_3_ using a microwave dissolver (CEM Mars 5, Matthews, NC, USA). After cooling, the digested solutions were brought to 25 mL with Milli-Q water. Zinc concentrations were determined by inductively coupled plasma–optical emission spectrometry (iCAP 6300, Thermo Fisher Scientific, Waltham, MA, USA). The proportion of zinc of each component to total grain zinc content was computed as the following formula:(1)$$P\left(\%\right)=\frac{SC_{c}\times P_{c}}{SC_{g}}\times 100$$
where $SC_{c}$ refers to the zinc content in each component of the rice grain, $P_{c}$ refers to the proportion of each component in the rice grain, $SC_{g}$ refers to the zinc content in the rice grain.

#### 2.3.4. Determination of the Phytic Acid Content and Molar Ratio of Phytic Acid to Zn

Phytic acid was quantified following a modified procedure [10]. A 0.25 g aliquot was extracted with dilute HCl, shaken, and centrifuged. The clarified supernatant was combined with FeCl_3_–sulfosalicylic acid chromogenic reagent and a sodium phytate standard. Absorbance at 500 nm was recorded, and phytic acid concentration was calculated. The molar ratio of phytate to zinc was derived by dividing millimoles of phytic acid by millimoles of zinc within the same sample.

#### 2.3.5. Rice Processing Quality and Appearance Quality

Post-harvest, panicles were sun-dried and grains equilibrated at ambient temperature for over 30 days. A 100 g subsample was de-husked; intact and broken kernels were separated and weighed. All quality traits were determined in accordance with GB/T 17891-2017. Brown rice rate, milled rice rate, and head rice rate were expressed as percentages of the original 100 g sample. Chalkiness grain rate and chalkiness degree were quantified with a rice appearance scanner (MRS-9600TFU2L, Shanghai, China).

#### 2.3.6. Rice Cooking Taste Quality

Rice palatability was quantified with a taste analyzer (Satake Corporation, Higashi-Hiroshima, Japan). An amount of 30 g of head rice was rinsed three times in 40 g of deionized water, soaked for 30 min, then steamed for 30 min under a filter paper. After 10 min warming, samples were fan-cooled by blowing air for 20 min and cooled naturally for 90 min at ambient temperature before analysis.

Rice-flour pasting was analyzed with a rapid viscosity analyzer (RVA, Super3, Newport Scientific, Warriewood, Australia). A 3 g sample (14% moisture basis) was dispersed in 25 g of distilled water in an aluminum canister. The RVA was programmed to hold the sample at 50 °C for 1 min, increase to 95 °C at 11.84 °C min^−1^, hold for 2.5 min, and then cool to 50 °C at 11.84 °C min^−1^. Peak viscosity, trough viscosity, final viscosity, breakdown value, setback value, and consistency viscosities were extracted from the resulting curves.

#### 2.3.7. Protein and Amylose Content in Rice

Protein content in milled rice was determined using the Kjeldahl method with an automatic Kjeldahl apparatus (Kjeltec 8200, FOSS, Hillerød, Denmark). Amylose content was measured using a Near-Infrared Grain Analyzer (Infratec 1241, FOSS, Denmark).

#### 2.3.8. Statistical Analysis

The data were presented as mean ± standard deviation (SD), and statistical analyses were conducted by LSD test using IBM SPSS Statistics software (Version 26.0; IBM Corp., Armonk, NY, USA). Two-way analysis of variance (ANOVA) with year and treatment was conducted to identify significant differences among treatment means, with significance set at *p* < 0.05.

## 3. Results

### 3.1. Grain Yield and Yield Components

Table 1 shows that, despite a pronounced inter-annual yield gap, as grain output in 2021 exceeded that of 2020, foliar spraying of ZnO NPs consistently outperformed the control across both years. Relative to the control, grain yield treated with ZnO NPs foliar application rose by 1.40–4.62%, with the T2 and T3 regimes accounting for the significant increments (*p* < 0.05). Dissection of yield components indicated that the gain was driven almost exclusively by a higher filled-grain rate (0.67–1.35%) and heavier 1000-grain weight (1.62–4.07%) as the ZnO NPs dose escalated. Panicle density and spikelets per panicle remained statistically unchanged across all treatments.

### 3.2. Net Photosynthetic Rate and SPAD Value

Table 2 reveals that, although flag-leaf net photosynthetic rate (*P_n_*) and SPAD values were significantly higher in 2021 than in 2020 (*p* < 0.05), foliar application of ZnO NPs consistently enhanced both parameters across the two years. At the heading stage, Pn was increased by 1.44–5.25% relative to the control, with the magnitude of increment rising to 2.05–5.31% at 20 days after heading and further to 2.62–6.90% at 40 days after heading as ZnO NPs dosage increased. SPAD values exhibited a comparable trend, with improvements of 1.84–3.94%, 1.90–4.29%, and 2.84–6.04% observed at the corresponding phenological stages. By maturity, the increments in *P_n_* and SPAD value had attenuated and were no longer statistically significant. The progressive enhancement from heading to 40 days after heading indicates a sustained, cumulative stimulation of photosynthetic capacity throughout grain filling.

### 3.3. Processing and Appearance Quality

As shown in Table 3, in 2021, the milled rice rate, head rice rate, chalky grain rate, and chalkiness degree were significantly higher than those in 2020, whereas the brown rice rate did not differ between the two years (*p* < 0.05). Foliar application of ZnO NPs improved processing quality parameters—including brown rice rate, milled rice rate, and head rice rate—in a dose-dependent manner. Specifically, the brown rice rate increased by 0.16–0.83%, with significant increments observed under the T2 and T3 treatments. Relative to the control (CK), the milled rice rate and head rice rate also exhibited significant improvements, by 0.33–1.16% and 0.59–1.61%, respectively. For appearance quality, ZnO NPs application markedly reduced the chalky grain rate and chalkiness degree by 5.73–18.62% and 11.57–27.18%, respectively, indicating a distinct enhancement in grain appearance with greater transparency.

### 3.4. Tasting and Cooking Qualities

As shown in Table 4, significant differences in amylose content and rice taste value were observed between the two years (*p* < 0.05). Additionally, the interaction between ZnO NPs treatment and year significantly affected rice taste value. With increasing ZnO NPs dosage, rice protein content exhibited an upward trend, whereas amylose content decreased. Although protein content showed slight increments (0.80–2.90%) across both years of treatment, ANOVA results indicated these changes were not statistically significant. In contrast, amylose content was significantly reduced by 3.72–6.76% under ZnO NPs treatments. Relative to the control (CK), rice taste value improved by 1.61–7.22% with escalating ZnO NPs dosage, indicating enhanced palatability. Component analysis of taste value revealed increases in appearance value (1.94–9.75%), viscosity value (2.80–7.47%), and balance value (1.73–7.73%), coupled with a significant reduction in hardness (2.81–13.79%); these changes collectively contributed to superior eating quality under ZnO NPs treatments.

In the RVA profile, significant differences in setback values and consistency values were observed between the two-year trials (*p* < 0.05). Foliar ZnO NPs treatment exerted a significant effect on peak viscosity, breakdown value, and setback values, with no significant interaction detected between ZnO NPs treatment and year. The improvements in rice taste were further evident in the RVA profile (Table 5): ZnO NPs application significantly increased the breakdown value by 5.36–15.63% and reduced the setback value by 8.98–24.53%, primarily attributed to an increase in peak viscosity (2.49–6.78%) relative to the control (CK), indicating superior rice taste with a softer, more elastic texture. No significant differences were observed in trough viscosity, final viscosity, or consistency value across treatments.

### 3.5. Zinc Content, Distribution, and Bioavailability

With respect to zinc content, distribution, and bioavailability within grains, significant differences were exclusively observed among the ZnO NPs treatments. In contrast, no significant differences were detected between years, nor was there a significant interaction between years and treatments. As illustrated in Figure 2, foliar application of ZnO NPs significantly increased zinc content in all rice grain fractions in a dose-dependent manner. Total grain zinc content was elevated by 21.00–73.49% relative to the control. In edible components, zinc content increased by 23.73–85.10% in brown rice and 29.07–103.07% in polished rice, with these increments being notably greater than those observed in non-edible fractions: glume (12.81–37.01%) and rice bran (9.71–36.77%). This induced a significant redistribution of zinc within the grain: the proportion of zinc allocated to brown rice and polished rice increased by 2.45–7.56% and 7.05–18.38%, respectively, whereas the proportions in glume and rice bran decreased by 7.66–24.46% and 10.26–22.32%, respectively. These findings indicate that foliar ZnO NPs application promotes preferential zinc accumulation in the edible portions of the grain. Additionally, the phytate-to-zinc molar ratio declined significantly with increasing ZnO NPs dosage (Figure 3), with reductions of 18.46–44.31% in brown rice and 21.67–48.39% in polished rice, thereby demonstrating enhanced zinc bioavailability in the edible grain fractions.

## 4. Discussion

### 4.1. Effects of ZnO NPs on Rice Yield and Grain Quality

Numerous studies have demonstrated that zinc plays a crucial role in mitigating the accumulation of reactive oxygen species (ROS) and delaying leaf senescence in rice plants. This is attributed to the fact that zinc is indispensable for enhancing the activity of antioxidant enzymes, such as peroxidase (POD), superoxide dismutase (SOD), and catalase (CAT) [32,33]. During the post-heading stage in rice, the rapid grain filling process is often accompanied by accelerated leaf aging, which may impair photosynthetic activity and the translocation of assimilates to developing grains [34]. Therefore, appropriate zinc supplementation during this critical stage is essential for improving grain filling and achieving higher yield [35]. In comparison with traditional zinc fertilizers, zinc oxide nanoparticles (ZnO NPs) exhibit a larger specific surface area, higher surface activity, as well as superior permeability—attributes stemming from their nanoscale dimension. These characteristics enable it to be absorbed and utilized by rice plants more rapidly and efficiently [36], particularly in the context of photosynthesis in rice leaves [37,38]. In this study, although there were certain variations in the relevant data of rice under ZnO NPs treatment across different years, overall, ZnO NPs consistently and effectively exerted significant positive effects on rice growth, yield, and quality. In particular, foliar application of ZnO NPs significantly improved both net photosynthetic rate and SPAD value in flag leaves from heading to maturity (a period critical for grain filling) [39]. The enhanced photosynthetic efficiency promoted a higher filled-grain rate and increased 1000-grain weight, ultimately contributing to the observed yield improvement. Moreover, the improved grain filling driven by ZnO NPs also positively influenced grain quality. On one hand, greater grain filling promotes the accumulation of storage materials, thereby increasing brown rice rate, polished rice rate, and head rice rate, indicators closely associated with rice processing quality [40]. On the other hand, enhanced grain filling contributes to a more compact starch arrangement and cellular structure, which helps reduce the formation of chalky areas in rice grains [41,42]. This was supported by the significant reductions in chalky grain rate and chalkiness degree observed in ZnO NP-treated rice.

With regard to eating quality (one of the most critical factors for consumer acceptance) [43], ZnO NP application significantly improved sensory attributes, including appearance value, viscosity value, and balance value, while reducing hardness. These improvements were further corroborated by the RVA (Rapid Visco Analyzer, Perten Instruments, Hägersten, Sweden) profile, where ZnO NP treatment increased peak viscosity and breakdown value and reduced setback value. Higher peak and breakdown values, along with a lower setback, typically indicate softer texture, better elasticity, and overall superior eating quality [44,45]. The primary reason for the improved palatability under ZnO NP treatment appears to be the reduction in amylose content. Previous studies have demonstrated that rice eating quality is negatively correlated with both protein and amylose content [46]. In this study, although protein content increased slightly, the change was not statistically significant. In contrast, the marked reduction in amylose content induced by ZnO NPs played a more direct and decisive role in enhancing rice texture and taste.

### 4.2. Effects of ZnO NPs on Zn Enrichment and Bioavailability

Foliar application of zinc fertilizer has been widely demonstrated to increase grain zinc content in rice [47]. Compared to conventional zinc formulations like ZnO and ZnSO_4_, zinc in nanoparticulate form, like ZnO NPs, exhibits enhanced foliar adhesion, superior mobility, and more efficient uptake and translocation within plant tissues [48,49,50,51]. Consequently, it can be effectively utilized as a zinc fertilizer to promote zinc fortification and enrichment in rice products [52]. Currently, the most prevalent approach for applying ZnO NPs in rice cultivation is foliar spraying during the mid-to-late growth stages. This method enables more effective zinc accumulation while requiring lower application rates [53,54]. Although a portion of the spray may drip onto the soil, ZnO NPs exhibit minimal toxicity to plants at low concentrations [55,56]. Furthermore, in crop rotation systems, this application method facilitates the absorption and utilization of zinc by subsequent crops [57,58]. Nevertheless, the risk of excessive zinc accumulation necessitates consideration when zinc is applied over the long term. Our results are consistent with these findings: The enhancing effect of ZnO NPs on grain zinc nutrition was consistent and reproducible across the two years; moreover, ZnO NPs significantly increased zinc concentrations in all parts of the rice grain, with particularly pronounced increases observed in the edible components (brown rice and polished rice) compared to non-edible tissues such as husk and bran [59]. Analysis of zinc distribution revealed that ZnO NP application effectively promoted zinc enrichment in edible grain portions. The proportion of zinc in brown rice and polished rice increased significantly, while that in the glume and rice bran decreased. This redistribution enhances the nutritional value of rice for human consumption. However, total zinc content alone does not determine zinc bioavailability. Phytic acid (phytate), an anti-nutritional factor commonly found in cereal grains, binds zinc to form insoluble complexes, reducing its intestinal absorption [60]. As such, the molar ratio of phytate to zinc is often used as an index of zinc bioavailability, with values below 15 considered acceptable for adequate absorption [61,62,63]. In this study, foliar ZnO NP application significantly reduced the phytate-to-zinc molar ratio in both brown rice and polished rice, indicating enhanced zinc bioavailability in both years. This improvement may be attributed to the mechanism of foliar fertilization, which, unlike soil-applied zinc, does not lead to antagonistic interactions with phosphorus in the soil [64]. Consequently, foliar application does not stimulate phytate synthesis in the grain, allowing for higher zinc retention in a more bioavailable form. These findings highlight the potential of ZnO NPs as an effective foliar biofortification strategy to address dietary zinc deficiency.

## 5. Conclusions

This study demonstrated that foliar application of ZnO NPs at the gestation stage significantly improved rice yield, grain quality, and zinc nutrition. Enhanced net photosynthetic rate and SPAD value contributed to better grain filling and increased yield. ZnO NPs also improved key eating quality traits by enhancing taste value and viscosity parameters while reducing chalkiness and amylose content. Additionally, ZnO NPs application effectively enhanced zinc concentration and its proportion in brown and polished rice, while decreasing the phytate-to-zinc molar ratio, thereby improving zinc bioavailability. Under the tested conditions in this study, these findings suggest that ZnO NPs are a promising foliar fertilizer for simultaneous yield enhancement, grain quality improvement, and micronutrient biofortification in rice production.

## Figures and Tables

**Figure 1 foods-14-03018-f001:**
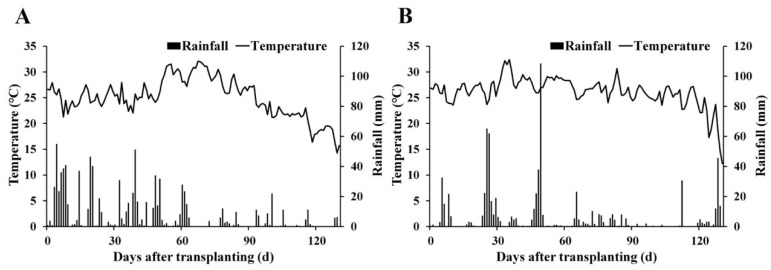
Rainfall and temperature from transplanting to harvest in 2020 (**A**) and 2021 (**B**).

**Figure 2 foods-14-03018-f002:**
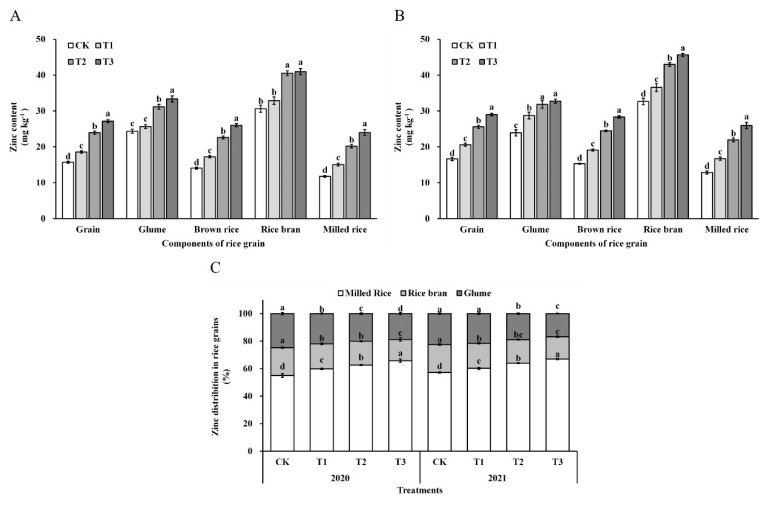
Zinc content (2020—(**A**) and 2021—(**B**)) and distribution (**C**) in rice grain. Error bars show standard error of replicates (*n* = 3). Values followed by different lowercase letters were significantly different at the 0.05 probability level among different treatments.

**Figure 3 foods-14-03018-f003:**
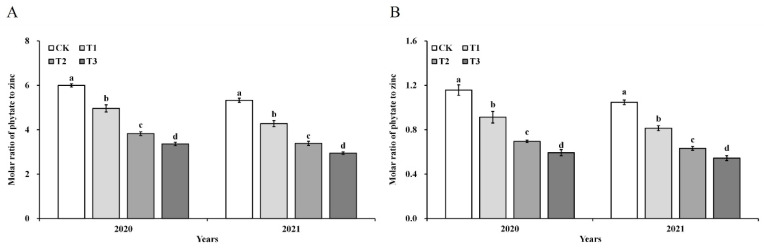
Molar ratio of phytate to zinc in brown rice (**A**) and polished rice (**B**). Error bars show standard error of replicates (*n* = 3). Values followed by different lowercase letters were significantly different at the 0.05 probability level among different treatments.

**Table 1 foods-14-03018-t001:** Effect of foliar spraying Fe fertilizers on yield and its components.

Year	Treatment	Panicles (×10^4^ hm^−2^)	Spikelets per Panicle	Filled-Grain Rate (%)	1000-Grain Weight (g)	Grain Yield (t hm^−2^)
2020	CK	327.44 ± 7.96 a	118.63 ± 3.51 a	91.54 ± 0.28 e	27.32 ± 0.10 f	9.85 ± 0.21 f
	T1	326.23 ± 9.17 a	118.07 ± 6.97 a	92.15 ± 0.30 cd	27.78 ± 0.14 de	9.98 ± 0.16 ef
	T2	329.07 ± 9.58 a	119.39 ± 4.78 a	92.56 ± 0.27 bc	28.18 ± 0.31 bc	10.16 ± 0.21 cd
	T3	327.96 ± 8.06 a	117.97 ± 6.28 a	92.83 ± 0.20 ab	28.43 ± 0.26 ab	10.31 ± 0.13 bc
2021	CK	326.76 ± 9.59 a	119.57 ± 3.54 a	91.99 ± 0.29 de	27.47 ± 0.10 ef	10.07 ± 0.14 de
	T1	327.34 ± 9.14 a	120.30 ± 6.02 a	92.62 ± 0.47 abc	27.90 ± 0.08 cd	10.22 ± 0.11 cd
	T2	325.92 ± 7.28 a	119.80 ± 3.15 a	93.00 ± 0.29 ab	28.24 ± 0.28 abc	10.40 ± 0.12 ab
	T3	328.06 ± 8.72 a	118.91 ± 4.00 a	93.19 ± 0.19 a	28.60 ± 0.11 a	10.52 ± 0.15 a
	Y	NS	NS	**	NS	**
	T	NS	NS	**	**	**
	Y × T	NS	NS	NS	NS	NS

Note: Different lowercase letters in the same column indicate significant difference of 5% (results in different years were compared, respectively). ** are significant differences at the 0.01 probability levels. NS: no significant difference.

**Table 2 foods-14-03018-t002:** Effects of foliar spraying Fe fertilizers on net photosynthetic rate and SPAD value after heading.

Year	Treatment	Net Photosynthetic Rate (μmol m^−2^ s^−1^)				SPAD Value			
		Heading	20 Days After Heading	40 Days After Heading	Maturity	Heading	20 Days After Heading	40 Days After Heading	Maturity
2020	CK	25.85 ± 0.14 e	20.16 ± 0.09 f	13.73 ± 0.12 e	7.35 ± 0.22 a	44.17 ± 0.59 e	34.83 ± 0.35 f	23.27 ± 0.32 f	10.77 ± 0.15 e
	T1	26.22 ± 0.17 d	20.55 ± 0.13 de	14.11 ± 0.18 cd	7.46 ± 0.21 a	45.00 ± 0.26 cd	35.50 ± 0.26 d	23.93 ± 0.15 d	10.83 ± 0.21 de
	T2	26.80 ± 0.20 b	20.92 ± 0.19 c	14.24 ± 0.13 cd	7.43 ± 0.12 a	45.50 ± 0.36 bc	36.00 ± 0.20 c	24.40 ± 0.20 c	10.90 ± 0.10 cde
	T3	27.25 ± 0.07 a	21.24 ± 0.18 ab	14.70 ± 0.18 ab	7.50 ± 0.22 a	46.17 ± 0.25 a	36.33 ± 0.15 b	24.70 ± 0.10 b	10.97 ± 0.12 bcde
2021	CK	26.14 ± 0.11 d	20.30 ± 0.14 ef	14.02 ± 0.15 d	7.57 ± 0.19 a	44.80 ± 0.20 d	35.17 ± 0.31 e	23.63 ± 0.15 e	11.03 ± 0.06 abcd
	T1	26.52 ± 0.18 c	20.74 ± 0.14 cd	14.37 ± 0.14 c	7.61 ± 0.12 a	45.60 ± 0.26 b	35.83 ± 0.40 c	24.30 ± 0.20 c	11.10 ± 0.10 abc
	T2	26.92 ± 0.18 b	21.01 ± 0.16 bc	14.63 ± 0.15 b	7.67 ± 0.16 a	46.03 ± 0.15 ab	36.33 ± 0.21 b	24.77 ± 0.15 b	11.17 ± 0.06 ab
	T3	27.47 ± 0.20 a	21.37 ± 0.16 a	14.96 ± 0.12 a	7.71 ± 0.25 a	46.30 ± 0.40 a	36.67 ± 0.15 a	25.03 ± 0.21 a	11.20 ± 0.10 a
	Y	**	*	**	**	**	**	**	**
	T	**	**	**	NS	**	**	**	**
	Y × T	NS	NS	NS	NS	NS	NS	NS	NS

Note: Different lowercase letters in the same column indicate significant difference of 5% (results in different years were compared, respectively). * and ** are significant differences at the 0.05 and 0.01 probability levels, respectively. NS: no significant difference.

**Table 3 foods-14-03018-t003:** Effects of foliar spraying Fe fertilizers on rice processing quality and appearance quality.

Year	Treatment	Brown Rice Rate (%)	Polished Rice Rate (%)	Head Rice Rate (%)	Chalkiness Grain Rate (%)	Chalkiness Degree (%)
2020	CK	83.98 ± 0.26 e	73.60 ± 0.16 e	60.33 ± 0.23 e	31.86 ± 1.39 a	11.40 ± 0.70 a
	T1	84.12 ± 0.16 de	73.83 ± 0.23 d	60.64 ± 0.29 d	30.15 ± 0.83 b	10.14 ± 0.35 b
	T2	84.52 ± 0.19 bc	74.29 ± 0.25 c	60.84 ± 0.14 d	27.31 ± 0.65 d	9.59 ± 0.42 b
	T3	84.64 ± 0.22 bc	74.44 ± 0.09 bc	61.15 ± 0.12 c	25.66 ± 0.44 e	8.69 ± 0.25 c
2021	CK	84.37 ± 0.15 cd	73.98 ± 0.22 d	60.71 ± 0.13 d	28.79 ± 0.92 c	9.45 ± 0.71 b
	T1	84.51 ± 0.18 bc	74.25 ± 0.14 c	61.11 ± 0.18 c	27.04 ± 0.38 d	8.31 ± 0.50 cd
	T2	84.77 ± 0.18 b	74.57 ± 0.18 b	61.43 ± 0.13 b	25.48 ± 0.76 e	7.80 ± 0.39 d
	T3	85.10 ± 0.10 a	74.86 ± 0.20 a	61.84 ± 0.22 a	23.67 ± 0.67 f	6.56 ± 0.44 e
	Y	NS	**	**	**	**
	T	**	**	**	**	**
	Y × T	NS	NS	NS	*	**

Note: Different lowercase letters in the same column indicate significant difference of 5% (results in different years were compared, respectively). * and ** are significant differences at the 0.05 and 0.01 probability levels, respectively. NS: no significant difference.

**Table 4 foods-14-03018-t004:** Effect of foliar spraying Fe fertilizers on rice nutritional quality and taste quality.

Year	Treatment	Protein Content (%)	Amylose Content (%)	Tasting Value	Appearance Value	Hardness Value	Viscosity Value	Balance Value
2020	CK	7.34 ± 0.10 ab	12.56 ± 0.12 a	82.80 ± 0.62 f	8.47 ± 0.06 f	6.00 ± 0.10 a	8.93 ± 0.06 e	8.57 ± 0.06 f
	T1	7.40 ± 0.10 ab	12.13 ± 0.11 b	83.97 ± 0.15 e	8.60 ± 0.13 e	5.83 ± 0.06 b	9.10 ± 0.10 d	8.67 ± 0.06 e
	T2	7.47 ± 0.06 ab	11.93 ± 0.12 c	87.53 ± 0.40 c	9.07 ± 0.06 c	5.33 ± 0.06 e	9.40 ± 0.10 b	9.17 ± 0.10 b
	T3	7.55 ± 0.14 a	11.70 ± 0.20 d	89.07 ± 0.25 b	9.30 ± 0.10 b	5.13 ± 0.06 f	9.63 ± 0.06 a	9.37 ± 0.06 a
2021	CK	7.27 ± 0.10 b	12.15 ± 0.16 b	84.43 ± 0.12 e	8.63 ± 0.06 e	5.83 ± 0.06 b	8.90 ± 0.10 e	8.70 ± 0.13 e
	T1	7.33 ± 0.11 b	11.66 ± 0.14 d	85.97 ± 0.06 d	8.83 ± 0.10 d	5.67 ± 0.06 c	9.23 ± 0.06 c	8.90 ± 0.06 d
	T2	7.45 ± 0.09 ab	11.50 ± 0.16 de	87.87 ± 0.31 c	9.23 ± 0.06 b	5.47 ± 0.06 d	9.30 ± 0.13 bc	9.03 ± 0.06 c
	T3	7.49 ± 0.09 ab	11.34 ± 0.12 e	90.23 ± 0.42 a	9.47 ± 0.13 a	5.07 ± 0.06 f	9.53 ± 0.06 a	9.23 ± 0.10 b
	Y	NS	**	**	**	*	NS	NS
	T	NS	**	**	**	**	**	**
	Y × T	NS	NS	**	NS	**	*	**

Note: Different lowercase letters in the same column indicate significant difference of 5% (results in different years were compared, respectively). * and ** are significant differences at the 0.05 and 0.01 probability levels, respectively. NS: no significant difference.

**Table 5 foods-14-03018-t005:** Effect of foliar spraying Fe fertilizers on RVA parameters.

Year	Treatment	Peak Viscosity (cP)	Trough Viscosity (cP)	Breakdown Value (cP)	Final Viscosity (cP)	Setback Value (cP)	Consistence Value (cP)
2020	CK	2884.00 ± 36.17 d	1703.67 ± 21.57 a	1180.33 ± 26.89 d	2283.33 ± 22.05 a	−600.67 ± 16.74 a	579.67 ± 22.51 abc
	T1	2959.00 ± 39.34 bcd	1702.67 ± 25.42 a	1256.33 ± 17.79 c	2297.33 ± 30.66 a	−661.67 ± 11.93 b	594.67 ± 17.01 ab
	T2	3001.33 ± 42.48 bc	1691.33 ± 16.01 a	1310.00 ± 26.91 b	2310.67 ± 44.99 a	−690.67 ± 12.52 bc	619.33 ± 29.40 a
	T3	3098.33 ± 41.67 a	1740.00 ± 17.58 a	1358.33 ± 40.82 a	2341.33 ± 31.21 a	−757.00 ± 14.73 d	601.33 ± 26.63 ab
2021	CK	2932.33 ± 55.59 cd	1730.67 ± 31.56 a	1201.67 ± 30.75 d	2274.33 ± 30.73 a	−658.00 ± 24.88 b	543.67 ± 13.58 c
	T1	3002.00 ± 23.52 bc	1749.00 ± 20.95 a	1253.00 ± 19.00 c	2292.67 ± 19.43 a	−709.33 ± 18.56 c	543.67 ± 17.52 c
	T2	3031.33 ± 41.79 ab	1724.00 ± 36.37 a	1307.33 ± 11.50 b	2284.33 ± 38.28 a	−747.00 ± 15.39 d	560.33 ± 14.53 bc
	T3	3112.00 ± 25.53 a	1716.00 ± 31.58 a	1396.00 ± 17.06 a	2302.33 ± 21.55 a	−809.67 ± 12.90 e	586.33 ± 29.02 abc
	Y	NS	NS	NS	NS	*	**
	T	**	NS	**	NS	**	NS
	Y × T	NS	NS	NS	NS	NS	NS

Note: Different lowercase letters in the same column indicate significant difference of 5% (results in different years were compared, respectively). * and ** are significant differences at the 0.05 and 0.01 probability levels, respectively. NS: no significant difference.

## Data Availability

The original contributions presented in the study are included in the article, further inquiries can be directed to the corresponding author.

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
