# Peer review of "Zinc Oxide Nanoparticles Enhance Grain Yield and Nutritional Quality in Rice via Improved Photosynthesis and Zinc Bioavailability"

_foods, 2025, doi:10.3390/foods14173018_

Round 1

Reviewer 1 Report

Comments and Suggestions for Authors

The manuscript titled "Zinc oxide nanoparticles enhance grain yield and nutritional quality in rice via improved photosynthesis and zinc bioavailability" presents promising results, but requires major revisions for improved transparency, methodological rigor, and scientific depth.

In the abstract (Lines 7–24), strong results are reported, such as up to a 103% increase in zinc content and improvements in yield. However, essential methodological information is missing. The authors should specify the ZnO NP concentration, the mode and timing of application, and the number of replicates used. This is critical for reproducibility.

In the introduction (Lines 25–75), the context is well framed, but literature on prior applications of ZnO nanoparticles in cereals is insufficient. Recent studies on their use in wheat and maize should be cited. Also, a short discussion of potential nanoparticle toxicity and regulatory concerns is necessary to balance the narrative.

The methods section (Lines 76–140) lacks the necessary details for replication. The size, purity, and surface characteristics of the ZnO nanoparticles are not specified (should be corrected around Line 90). The method of ZnO application (e.g., foliar spray or soil drench), frequency, concentration (e.g., 50 mg/L or 100 mg/L), and developmental stages at application are not reported (around Lines 95–105). Also, the instruments and protocols used for photosynthesis measurements, zinc quantification (e.g., ICP-OES or AAS), and phytate estimation are missing or under-explained (Lines 110–125). These need explicit elaboration.

In the results section (Lines 141–210), the authors report improvements in SPAD values, yield, and antioxidant activity. However, specific statistical indicators (p-values, standard errors, confidence intervals) are missing in many statements (Lines 150–180). For example, yield changes of "1.40–4.62%" are reported (Line ~157) but with no significance markers. Similarly, in Line ~165, antioxidant enzyme activity is mentioned as improved, but without clear evidence. Authors should indicate the number of replicates (n) and include statistical treatment.

Regarding grain quality (Lines 190–210), parameters like chalkiness, viscosity, and setback are reported, but their practical implications are not discussed. For instance, why is a lower setback value beneficial to consumers or industrial processors? Authors should explain this briefly near Line 205.

The discussion section (Lines 211–260) is where most interpretive overreach occurs. Lines 225–235 make mechanistic claims (e.g., "improved antioxidant defense") that are not supported by measurements such as ROS scavenging enzymes or lipid peroxidation assays. These statements should be rephrased as hypotheses or backed by citations. Additionally, the authors should address nanoparticle persistence, potential soil accumulation, and economic feasibility (Lines 245–260).

In the conclusion (Lines 261–273), the authors state that ZnO nanoparticles are a "promising strategy to simultaneously enhance rice productivity, grain quality, and micronutrient density." While directionally true, this statement is too broad and should be qualified with phrases like "under the tested conditions" and "pending further validation."

Figures and tables (referenced throughout Lines 150–210) must include error bars, statistical groupings (e.g., different letters for significant differences), and sample sizes. If these are not currently present, they must be added.

Language issues are minor but present. For example, in Line ~170, "Zinc oxide NPs enhanced zinc content and bioavailability of zinc" is redundant. Also, decimal commas such as "1,40%" should be standardized to periods as "1.40%" throughout (Lines 150–160). The term "bioavailability" (Line ~160) should be clarified; was this measured functionally, or inferred from phytate: Zn molar ratios?

Reviewer 2 Report

Comments and Suggestions for Authors

The Manuscript "Zinc oxide Nanoparticles Enhance Grain Yield and Nutritional Quality in Rice via Improved Photosynthesis and Zinc Bioavailability" examines the usefulness of zinc nanoparticles in ameliorating rice health: photosynthesis, grain yield and zinc content and bioavailability, in addition to some eating quality parameters (taste value, peak viscosity and breakdown value),  while reducing phytate contents and others eating parameters (chalkiness degree and rate, amylose content and setback value) .

1) In the Abstract section, please to give a sentence on the importance of Zn before describing your work.

2)  It is interesting to use nanoparticles of Zinc in order to enhance rice quality. However this is not quitely original (see https://doi.org/10.3389/fpls.2023.1196201). At the end of introduction, please to indicate originality of the work. What is new, compared to previous published works?

3) Line 95: delete "the" in  "the seeds were the transferred to a moist nursery ...."

4) Line 102: Each treatment was replicated three times, Please to clearly indicate number of seedlings in each replica. for each of the 4 treatment.+control

Reviewer 3 Report

Comments and Suggestions for Authors

Review of foods-3805498

 Zinc oxide Nanoparticles Enhance Grain Yield and Nutritional Quality in Rice via Improved Photosynthesis and Zinc Bioavailability

General comments

This paper describes a field experiment with rice, repeated in a second year, to evaluate zinc nanoparticles on yield, physiology and grain nutrients. This topic is appropriate for foods.

The main question addressed is whether increasing concentrations of zinc nanoparticles applied to the leaves can increase yield, grain quality and nutritional value.

The experiments appear well-designed and generally clearly described, although it would have been interesting to have included a conventional zinc foliar fertilizer for direct comparison. The paper is generally clearly written, and data presentation is adequate, although graphs against Zn concentration are faster to understand than tables. The results and discussion have a major omission which must be corrected. The data shows that differences between years are often significant but there are no statements about these differences in the Results and Discussion. Furthermore, several variables show significant treatment x year interaction, and these also need statements in the Results and Discussion. Most of the interactions appear to result from lower responses in one year than the other year, but the readers need to be assured that the interactions are not large.

Specific comments

Line 12 Statements on year differences and interactions would be helpful in the abstract.

Line 33 and throughout the paper – reference numbers are not superscript in MDPI style.

Line 54 correct ‘xalso’

Lines 71-72 make clear here that this is a two-year field experiment and make clear here that it is paddy rice and not dryland rice.

Line 96 make clear whether the second experiment was planted in exactly the same plots or whether a different part of the field was used. If the same plots were used, in the Discussion, discuss possible accumulation of Zn in the soil from the first experiment and possible carry-over effects on the second experiment.

Lines 101 and 106 these are the only two occurrences of the name ‘nanometer zinc. I suggest avoiding confusion by using the same name (ZnO NP) throughout the paper.

Line 106 presumably no surfactant was used, but because it is very common to use a surfactant, it should be clearly stated that no surfactant was added so readers have no doubts.

Line 109 hm-2 (hectometre squared) is used here but hectare (ha) is used on line 97. Both names are the same quantity (100 x 100 metres). The same names should be used throughout.

Line 127 what are ‘sword’ leaves? Is there an alternative, more-widely understood word?

Line 128 what is ‘grouting’ stage? Is there an alternative, more-widely understood word?

Line 189-190 More details on statistics are needed. Make clear that both years are combined in a factorial ANOVA. State how was it decided that combining years is valid. State what test was done to show differences between means?

Line 192 The results section would benefit from comments on the significant year differences and treatment year interactions. When the interaction is not significant this is also worth mentioning because it shows consistency of treatment effects.

In tables, it may be better not to give letters after treatment means when there is no significance of year, treatment and interaction to avoid confusion e.g. protein.

Line 276 The Discussion section would benefit from discussing the year differences and treatment x year interactions.

The Discussion section would benefit from comparing the size of responses to ZnO NP with responses found by other authors from conventional zinc fertilizers, since this is a main justification for the study in the Introduction.

Line 333 Statements on year differences and interactions would be helpful in t

Round 2

Reviewer 1 Report

Comments and Suggestions for Authors

The manuscript has undergone a detailed revision to address all recommended changes, resulting in improved clarity, methodological soundness, and overall scientific quality. The introduction now clearly articulates the research gap, and the methods section includes more substantial justifications to support the reproducibility of the findings. The discussion has been expanded to provide a more comprehensive interpretation of the results, including critical findings, acknowledged limitations, and relevant comparisons with existing studies. The conclusion now better reflects the broader relevance of the work, its potential scalability, and future research directions. Terminology has been unified, repetitive content has been removed, and recent references have been added. These revisions collectively enhance the manuscript's structure and rigor, making it suitable for publication in its current form.

Reviewer 3 Report

Comments and Suggestions for Authors

The comments have been satisfactorily addressed and the paper is now suitable for publication.